# 7-*O*-tyrosyl Silybin Derivatives as a Novel Set of Anti-Prostate Cancer Compounds

**DOI:** 10.3390/antiox13040418

**Published:** 2024-03-29

**Authors:** Valeria Romanucci, Rita Pagano, Kushal Kandhari, Armando Zarrelli, Maria Petrone, Chapla Agarwal, Rajesh Agarwal, Giovanni Di Fabio

**Affiliations:** 1Department of Chemical Sciences, University of Naples “Federico II”, Complesso Monte Sant’Angelo, Via Cintia 4, I-80126 Napoli, Italy; valeria.romanucci@unina.it (V.R.); rita.pagano@unina.it (R.P.); zarrelli@unina.it (A.Z.); maria.petrone@unina.it (M.P.); 2Department of Pharmaceutical Sciences, Skaggs School of Pharmacy and Pharmaceutical Sciences, University of Colorado Anschutz Medical Campus, Aurora, CO 80045, USA; kushal.kandhari@cuanschutz.edu (K.K.); chapla.agarwal@cuanschutz.edu (C.A.); rajesh.agarwal@cuanschutz.edu (R.A.)

**Keywords:** silibinin, silybin, natural products, Mitsunobu reaction, antioxidants, human prostate cancer

## Abstract

Silybin is a natural compound extensively studied for its hepatoprotective, neuroprotective and anticancer properties. Envisioning the enhancement of silybin potential by suitable modifications in its chemical structure, here, a series of new 7-*O*-alkyl silybins derivatives were synthesized by the Mitsunobu reaction starting from the silybins and tyrosol-based phenols, such as tyrosol (TYR, **3**), 3-methoxytyrosol (MTYR, **4**), and 3-hydroxytyrosol (HTYR, **5**). This research sought to explore the antioxidant and anticancer properties of eighteen new derivatives and their mechanisms. In particular, the antioxidant properties of new derivatives outlined by the DPPH assay showed a very pronounced activity depending on the tyrosyl moiety (HTYR > MTYR >> TYR). A significant contribution of the HTYR moiety was observed for silybins and 2,3-dehydro-silybin-based derivatives. According to the very potent antioxidant activity, 2,3-dehydro-silybin derivatives **15ab**, **15a**, and **15b** exerted the most potent anticancer activity in human prostate cancer PC-3 cells. Furthermore, flow cytometric analysis for cell cycle and apoptosis revealed that **15ab**, **15a**, and **15b** induce strong G1 phase arrest and increase late apoptotic population in PC-3 cells. Additionally, Western blotting for apoptotic marker cleaved caspase-3 confirmed apoptosis induction by these silybin derivatives in PC-3 cells. These findings hold significant importance in the investigation of anticancer properties of silybin derivatives and strongly encourage swift investigation in pre-clinical models and clinical trials.

## 1. Introduction

Natural products of plant origin, and in particular polyphenols, have been the cornerstone of medicine since the first humans used them in their diet [1,2]. Recently, thanks to advances in drug screening techniques and the elucidation of many mechanisms of action, there has been a vigorous revival of interest in natural products for drug discovery [3]. One of the most attractive properties of polyphenols is their ability to scavenge radical species whose balance is vital for the normal functioning of the cell [4,5]. In fact, excessive ROS concentrations can induce cytotoxic events caused by protein, lipid, and DNA damage through so-called “oxidative stress”, which is the consequence of an imbalance between pro- and antioxidant species and is currently ascertained to be involved in aging and general inflammation [6,7,8].

Over the years, many studies have demonstrated that oxidative stress is involved in a wide range of pathologies including cancer. In this frame, ROS species seem to exert a critical role in tumorigenesis and cancer development. ROS are not only mediators of oxidative stress; for example, ROS-mediated DNA damage causes the malignant transformation of cells and promotes cancer initiation [9]. Several molecular features should be included in a rational drug design that aims at an amelioration of the pharmacological profile, and among them, antioxidant activity has to be taken into account. Many natural products with good radical scavenger properties include a large set of structures with a high degree of diversity and useful lead compounds for drug design [10].

Recently, our interest has focused on silibinin, a metabolite with a wide variety of pharmacological properties. Silibinin (Sil, **1ab**, Figure 1) is the major biologically active component of silymarin, an extract from the seeds of the milk thistle [*Silybum marianum* (L.) Gaertn.] [11]. Structurally, natural silibinin is a diastereoisomeric mixture of two flavonolignans, namely silybin A (SilA, **1a**) and silybin B (SilB, **1b**), in a ratio of approximately 1:1 (Figure 1) [12].

Silibinin, as well as other flavonolignans of silymarin, are considered as pleiotropic acting compounds since they are not only able to act as antioxidants but in specific conditions also as pro-oxidant agents. The discrepancy between synergy or antagonism in terms of antioxidant and pro-oxidant effects is well documented [13,14,15]. The structural motifs (as the eterocyclic ring C; the aromatic rings A, B, and E; the corresponding functions 7-, 5-, 3-, and 4″-OH; the carbonyl in position 4, and then the double bond C2-3 in the oxidized derivatives (DHS, Figure 1)) [16] play a key role in the determination of the radical scavenging activity and in the free radical interaction mechanism. Given the presence of different structural motifs that influence the antioxidant and pro-oxidant action, silibinin (Sil, **1ab**) does not present a high antioxidant activity, but nevertheless, many reports suggest that it has manifold inhibitory effects against various cancer cells including growth inhibition, anti-inflammation, cell cycle regulation, apoptosis induction, inhibition of angiogenesis, invasion, and metastasis [17]. In vitro cell-based and in vivo animal studies have demonstrated that Sil (**1ab**), as natural mixture of SilA and SilB, is a roaring lead compound for the design of new drugs for the treatment of prostate cancer. Silibinin has also been tested in prostate cancer patients in phase I–II pilot clinical trials, where it was reported to be well tolerated and showed plasma and target-tissue bioavailability [18]. However, both flavonolignans, Sil and DHS (Figure 1), suffer from poor oral bioavailability due to both the first-pass metabolism of glucuronidation and sulfation and low water solubility [19,20,21].

Different studies indicated that the in vitro antiproliferative activity of Sil and DHS against different prostate cancer cell lines can be further improved through appropriate chemical modifications. Particular attention has been addressed to the modifications on the phenolic hydroxyl group at C-7 with the aim of improving antioxidant activity, bioavailability, and anti-prostatic activity [22,23,24,25,26,27]. It must be said, however, that many studies both in vitro and in vivo clearly neglect the structure–activity relationship (SAR) of isomers, using silibinin as a natural mixture [28]. The understanding of the structural and functional properties’ impact of the flavonolignan core on pharmacological activity has recently led to more complete and detailed studies. Investigations on the role of the silybins hydroxy groups in relation to radical scavenging properties have highlighted that the number of hydroxyl substitutions in the backbone structure can affect both its pro-oxidant and antioxidant properties. Modification of the hydroxyl groups has also been found to affect the anticancer activity of polyphenols, and in detail, the 7-OH-group was found to exert a pro-oxidant activity in silibinin [29].

In this frame, our previous studies on the stereochemistry–activity relationship of silybins (SilA and SilB) revealed their features involved in the recognition of targets in AD and T2DM [30,31] as well as their ability to increase the proteasome activity [32], a multi-attractive target involved in protein conformational diseases [33,34]. Without neglecting the stereochemical aspect and aiming at improving the pharmacokinetics of silybins, in the last few years, we have also designed new silybin *pro*-drugs containing hydrophilic moieties which did not significantly modify silybin activities [35,36].

## 2. Materials and Methods

### 2.1. General

Silibinin **1** was purchased from Sigma-Aldrich (Milano, Italy). HPLC-grade ACN and CH_3_OH were purchased from Carlo Erba Reagents and Sigma-Aldrich. Unless otherwise indicated, other chemicals were obtained from Sigma Aldrich. For the experimental synthesis procedures of tyrosol-based building blocks **6**–**8**, see Romanucci et al. 2021 [37]. It must be noted that the experimental procedures for the synthesis of compounds **10**–**12** and **13**–**15** were described in detail only for the stereoisomer of silybin A (SilA, **1a**): the same reaction conditions (temperature, stoichiometric ratios, and time of reaction) were used for silybin B (SilB, **1b**) and silibinin (Sil, **1ab**) as a natural mixture. Reactions were monitored by TLC (precoated silica gel plate F254, Merck, Upper Gwynedd, PA, USA) and column chromatography: Merck Kieselgel 60 (70–230 mesh). The analysis was performed with a Shimadzu (Tokyo, Japan) LC-8A HPLC system equipped with a Shimadzu SCL-10A VP System control and Shimadzu SPD-10A VP UV-VIS Detector. HPLC purifications were carried out on a Phenomenex Gemini RP18 column (10 μm particle size, 21.20 mm × 250 mm i.d.) using a linear gradient of ACN in 0.1 M NH_4_Ac in H_2_O, pH 7.0, from 20% to 100% over 30 min at a flow rate of 8 mL/min, with detection at 288 and 260 nm. MALDI spectral data were acquired on a MALDI TOFTOF AB Sciex 5800 mass spectrometer.

### 2.2. Synthesis of 9″-O-tert-butyldimethylsilyl-silybin A (***9a***) and 9″-O-tert-buthyldimethylsilyl-silybin B (***9b***)

SilA **1** (1.0 g, 2.0 mmol) previously co-evaporated with anhydrous THF (three times) was dissolved in 12 mL of anhydrous ACN/DMF 3:1 (*v*/*v*) and 760 µL (7.0 mmol) of dry pyridine was added. The mixture was cooled to 0 °C and then TBDMSCl (630 mg, 4.2 mmol) was added and stirred. After 1 h (always at 0 °C), the disappearance of SilA by TLC control was observed, and about 1 mL of CH_3_OH was added and kept under stirring for a further 30 min. The mixture was diluted with DCM (100 mL) and was washed three times with saturated NaHCO_3_ solution. The organic phase was dried over anhydrous Na_2_SO_4_ and concentrated under reduced pressure. The crude material was purified by column chromatography eluted with DCM/ CH_3_OH 97:3 (*v*/*v*) and the derivative 9″-*O*TBDMS (**9a**) was obtained in good yield (1.1 g, 18.8 mmol, 74%).

(**9a**) R*_f_* = 0.5 (DCM-CH_3_OH 95:5, *v*/*v*); ^1^H NMR (CDCl_3_, 400 MHz): δ = 7.20 (*d*, *J* = 1.45 Hz, 1H, H-2′); 7.07 (*dd*, *J* = 8.44, 1.71 Hz, 1H, H-6′); 7.02 (*d*, *J* = 8.26 Hz, 1H, H-5′); 6.96 (*s*, 3H, H-2″, H-5″ and H-6″); 6.09 (*s*, 1H, H-8); 6.02 (*s*, 1H, H-6); 5.01–4.97 (overlapped signals, 2H, H-2, and H-7″); 4.55 (*d*, *J* = 11.8 Hz, 1H, H-3); 3.97 (*m*, 1H, H-8″); 3.91 (*s*, 3H, OCH_3_); 3.87 (*dd*, *J* = 11.9, 1.87 Hz, 1H, H-9″a); 3.57 (*dd*, *J* = 11.9, 2.84 Hz, 1H, H-9″b); 0.92 (*s*, 9H, (CH_3_)_3_CSiOCH_2_); 0.08 (*d*, *J* = 4.79 Hz, 6H, (CH_3_)_2_ SiOCH_2_) ppm. ^13^C NMR (CDCl_3_, 125 MHz): δ = 195.5; 167.5; 163.7; 163.1; 146.7; 146.1; 144.7; 143.9; 129.0; 128.4; 121.2; 120.8; 117.1; 116.3; 114.5; 109.8; 100.2; 97.2; 96.1; 83.0; 78.6; 76.2; 72.2; 62.3; 56.0; 25.8; 18.3; −5.12; −5.42 ppm. MS (MALDI-TOF) (+): calculated for [M] (C_31_H_36_O_10_Si) 596.70; found 597.55 [M + H]^+^, 619.86 [M + Na]^+^.

(**9b**) R*_f_* = 0.5 (DCM- CH_3_OH 95:5, *v*/*v*); ^1^H NMR (CDCl_3_, 400 MHz): δ = 7.17 (*d*, *J* = 1.84 Hz, 1H, H-2′); 7.08 (*dd*, *J* = 8.43, 1.90 Hz, 1H, H-6′); 7.01 (*d*, *J* = 8.23 Hz, 1H, H-5′); 6.96–6.94 (overlapped signals, 3H, H-2″, H-5″, and H-6″); 6.08 (*d*, *J* = 2.0 Hz, 1H, H-8); 6.03 (*d*, *J* = 2.0 Hz, 1H, H-6); 5.01–4.98 (overlapped signals, 2H, H-2, and H-7″); 4.55 (*d*, *J* = 11.8 Hz, 1H, H-3); 3.96 (*m*, 1H, H-8″); 3.88–3.85 (overlapped signals, 4H, OCH_3_, and H-9″a); 3.56 (*dd*, *J* = 11.8, 2.84 Hz, 1H, H-9″b); 0.91 (*s*, 9H, (CH_3_)_3_CSiOCH_2_); 0.07 (*d*, *J* = 4.98 Hz, 6H, (CH_3_)_2_ SiOCH_2_) ppm. ^13^C NMR (CDCl_3_, 125 MHz): δ = 195.7; 168.3; 163.8; 163.0; 147.2; 146.6; 144.5; 143.9; 129.2; 128.1; 120.8; 120.7; 117.2; 116.5; 114.9; 110.1; 100.2; 97.2; 96.1; 83.1; 78.6; 76.2; 72.3; 62.3; 55.9; 25.8; 18.3; −5.14; −5.44 ppm. MS (MALDI-TOF) (+): calculated for [M] (C_31_H_36_O_10_Si) 596.70; found 597.63 [M + H]^+^, 619.78 [M + Na]^+^.

### 2.3. Alkylation by Mitsunobu Reaction: General Procedure for the Synthesis of ***10a***–***12a***

To a solution of Ph_3_P and DIAD (0.50 mmol of each) dissolved in 2 mL of anhydrous THF and kept at 0 °C for 20 min, a mixture of 9″-O-protect silybin A **9a** (200 mg, 0.33 mmol) and tyrosyl building block (**6** or **7** or **8**, 0.33 mmol) was sequentially added and the resulting mixture was left to stir at 0 °C for 2 h. The reaction was monitored by TLC (hexane/AcOEt 85:15, *v*/*v*) and the crude reaction mixtures were found to be complex to purify, so after several attempts at simplification by crystallization, it was decided to carry out a coarse chromatography on the silica gel column eluted with hexane/AcOEt (80:20, *v*/*v*) and then to treat the crude fractions, previously dried under reduced pressure, with Et_3_N∙3HF (65 µL, 0.40 mmol) in 500 µL THF at rt. After silica gel chromatography eluted with DCM/ CH_3_OH (95:5, *v*/*v*), all compounds **10a**–**12a**, obtained in good yields (26–35%, see Table 1), were purified by RP-HPLC using a Phenomenex Gemini RP18 column (10 μm particle size, 21.20 mm × 250 mm i.d.) with a linear gradient of ACN in 0.1 M NH_4_OAc in H_2_O, pH 7.0, from 20% to 100% over 30 min at a flow rate of 8 mL/min, with detection at 288 and 260 nm. The identity of compounds **10a**–**12a** was confirmed by 1D and 2D NMR (^1^H and ^13^C) and MALDI-TOF analyses. The complete assignment was proved by COSY, HMBC, and HSQC NMR experiments (for signal assignment of **10a**–**12a**, see Table 2).

(**10b**) HPLC purity ≥ 99.0%; ^1^H-NMR (CD_3_OD, 500 MHz, 25 °C, δ ppm, *J* Hz); 7.13 (s, 1H, H2′); 7.10 (d, *J* = 8.1, 2H, H2‴, H6‴); 7.06 (dd, *J* = 8.3, *J* = 1.8, 1H, H6′); 7.04–7.00 (overlapped signals, 2H, H5′, H2″); 6.92 (dd, *J* = 8.2, *J* = 1.4, 1H, H6″); 6.86 (d, *J* = 8.2, 1H, H5″); 6.73 (d, *J* = 8.1, 2H, H5‴, H2‴); 6.08 (d, *J* = 1.8, 1H, H6); 6.04 (d, *J* = 1.8, 1H, H8); 5.02 (d, *J* = 11.4, 1H, H2); 4.94 (d, *J* = 8.0, 1H, H7″); 4.56 (d, *J* = 11.4, 1H, H3); 4.16 (t, *J* = 6.7, 2H, H8‴); 4.12–4.06 (m, 1H, H8″); 3.89 (s, 1H, OCH_3_); 3.74 (dd, *J* = 12.4, *J* = 2.2, 1H, H9″a); 3.51 (dd, *J* = 12.3, *J* = 4.4, 1H, H9″b); 2.96 (t, *J* = 6.7, 2H, H7‴) ppm. ^13^C-NMR (CD_3_OD, 125 MHz, 25 °C, δ ppm): 197.4; 167.6; 163.6; 162.8; 155.7; 148.0; 147.8; 146.9; 143.7; 130.0; 129.5 (×3); 128.6; 128.0; 120.8; 120.3; 116.4; 116.1; 114.8 (×2); 110.6; 101.1; 95.2; 94.1; 83.4; 78.6; 78.3; 72.3; 69.3; 60.6; 55.0; 34.1 ppm. MS (MALDI-TOF) (+): calculated for [M] (C_33_H_30_O_11_) 602.18; found 603.25 [M + H]^+^, 625.84 [M + Na]^+^.

(**10ab**) HPLC purity ≥ 99.0%; ^1^H-NMR (400 MHz, CD_3_OD, 25 °C, δ ppm, *J* Hz); 7.15–7.07 (overlapped signals, 3H, H2′, H2‴, H6‴); 7.07–6.99 (overlapped signals, 3H, H6′, H5′, H2″); 6.92 (dd, *J* = 8.1, *J* = 1.7, 1H, H6″); 6.85 (d, *J* = 8.0, 1H, H5″); 6.73 (d, *J* = 8.1, 2H, H5‴, H2‴); 6.07 (d, *J* = 2.0, 1H, H6); 6.03 (d, *J* = 2.0, 1H, H8); 5.01 (d, *J* = 11.5, 1H, H2); 4.93 (d, *J* = 8.1, 1H, H7″); 4.55 (d, *J* = 11.5, 1H, H3); 4.15 (t, *J* = 6.4, 2H, H8‴); 4.11–4.05 (m, 1H, H8″); 3.88 (s, 1H, OCH_3_); 3.72 (dd, *J* = 12.4, *J* = 2.4, 1H, H9″a); 3.50 (dd, *J* = 12.4, *J* = 4.5, 1H, H9″b); 2.95 (t, *J* = 6.4, 2H, H7‴) ppm. ^13^C NMR (100 MHz, DMSO-*d6*, 25 °C, δ ppm): 197.3; 167.6; 163.6; 162.8; 155.7; 148.0); 147.8; 144.1; 143.7; 130.1; 129.6 (×2); 129.0; 127.9; 121.6; 120.8; 120.3; 116.2; 116.3; 114.8 (×2); 110.6; 101.1; 95.1; 94.0; 83.3; 78.2; 76.2; 72.3; 69.3; 60.6; 54.9; 34.1 ppm. MS (MALDI-TOF) (+): calculated for [M] (C_33_H_30_O_11_) 602.18; found 603.66 [M + H]^+^, 625.48 [M + Na]^+^.

(**11b**) HPLC purity ≥ 99.0%; ^1^H-NMR (400 MHz, DMSO-*d6*, 25 °C, δ ppm, *J* Hz); 7.07 (s, 1H, H2′); 7.05–6.96 (overlapped signals, 3H, H5′, H6′, H2″), 6.90–6.83 (overlapped signals, 2H, H6″, H2‴); 6.81 (d, *J* = 8.2, 1H, H5″) 6.71–6.64 (overlapped signals, 2H, H-6‴, H5‴); 6.13–6.06 (overlapped signals, 2H, H6, H8); 5.11 (d, *J* = 11.1, 1H, H2); 4.90 (d, *J* = 7.8, 1H, H7″); 4.62 (d, *J* = 11.1, 1H, H3); 4.24–4.12 (overlapped signals, 3H, H8″, H8‴); 3.77 (s, 1H, OCH_3_); 3.73 (s, 1H, OCH_3_); 3.66–3.43 (overlapped signals with H_2_O, 1H, H9″a); 3.34 (dd, *J* = 12.5, *J* = 4.9, 1H, H9″b); 2.89 (t, *J* = 6.4, 2H, H7‴) ppm. ^13^C-NMR (100 MHz, DMSO-*d6*, 25 °C, δ ppm): 198.5; 167.2; 165.1; 162.7; 147.9; 147.6; 146.9; 144.7; 143.9; 143.5; 130.0; 129.6; 127.9; 121.8; 121.5; 120.9; 117.1; 116.8; 115.6, 115.5; 113.3; 111.7; 101.8; 95.8; 82.9; 78.2; 76.1; 71.8; 69.6; 60.4; 56.0; 55.8; 34.4 ppm. MS (MALDI-TOF) (+): calculated for [M] (C_34_H_32_O_12_) 632.19; found 633.68 [M + H]^+^, 655.56 [M + Na]^+^.

(**11ab**) HPLC purity ≥ 99.0%; ^1^H-NMR (400 MHz, CD_3_OD, 25 °C, δ ppm, *J* Hz, mixture of diastereoisomers); 7.10 (m, 1H, H6′); 7.07–6.98 (complex signals, 3H, H2′, H5‴, H2″), 6.90 (dd, *J* = 8.2, *J* = 1.6, 1H, H6″); 6.87–6.74 (complex signals, 2H, H-6T, H5′); 6.76–6.65 (complex signals, 2H, H5″, H2‴); 6.06 (dd, *J* = 1.8, 1H, H6); 6.01 (dd, *J* = 1.8, 1H, H8); 4.98 (d, *J* = 11.6, 1H, H2); 4.90 (overlapped signal with H_2_O, 1H, H7″); 4.51 (d, *J* = 11.5, 1H, H3); 4.14 (t, *J* = 6.9, 2H, H8‴); 4.06 (m, 1H, H8″); 3.87 (s, 1H, OCH_3_); 3.82 (s, 1H, OCH_3_); 3.71 (dd, *J* = 12.3, *J* = 2.3, 1H, -9″a); 3.49 (dd, *J* = 12.3, *J* = 4.6, 1H, H9″b); 2.95 (t, *J* = 6.2, 2H, H7‴) ppm. ^13^C-NMR (125 MHz, CD_3_OD, 25 °C, δ ppm): 197.3; 167.6; 163.5; 162.7; 147.8; 147.5; 146.9; 144.7; 144.1; 143.7; 130.0; 129.3; 128.0; 121.1; 120.8; 120.7; 120.2; 116.4 (x2); 116.2; 116.1; 114.8 (×2), 112.3; 110.6 (×2); 101.1; 95.2; 94.1; 83.3 (×2); 78.6; 76.2; 72.3; 69.3; 60.6; 55.0 (×2); 34.5 ppm. MS (MALDI-TOF) (+): calculated for [M] (C_34_H_32_O_12_) 632.19; found 633.25 [M + H]^+^, 655.56 [M + Na]^+^, 671.56 [M + K]^+^.

(**12b**) HPLC purity ≥ 99.0%; ^1^H-NMR (500 MHz, CD_3_OD, 25 °C, δ ppm, *J* Hz); 7.11 (s, 1H, H2′); 7.06–6.95 (overlapped signals, 3H, H6′, H5′, H2″), 6.90 (complex signal, 1H, H6″); 6.85 (d, *J* = 8.2, 1H, H5″); 6.73–6.67 (overlapped signals, 2H, H5‴, H2‴); 6.57 (d, *J* = 8.0, 1H, H6‴); 6.05 (s, 1H, H6); 6.00 (s, 1H, H8); 4.98 (complex signal, 1H, H2); 4.90 (overlapped signal with H_2_O, 1H, H7″); 4.50 (complex signal, 1H, H3); 4.16–4.00 (overlapped signals, 3H, H8‴, H8″); 3.87 (s, 1H, OCH_3_); 3.71 (d, *J* = 11.8, 1H, H9″a); 3.49 (dd, *J* = 12.3, *J* = 4.2, 1H, H9″b); 2.87 (s, 2H, H7‴) ppm. ^13^C-NMR (125 MHz, CD_3_OD, 25 °C, δ ppm): 197.3; 167.6; 163.5; 162.7; 147.8; 146.9; 144.8; 144.1; 143.7; 143.5; 130.0; 129.3; 128.0; 120.8; 120.3; 119.9; 116.4; 116.2 (×2); 115.7; 115.0; 114.9; 110.6; 101.1; 95.2; 94.1; 83.3; 78.6; 76.2; 72.3; 69.3; 60.6; 55.0; 34.3 ppm. MS (MALDI-TOF) (+): calculated for [M] (C_33_H_30_O_12_) 618.17; found 618.69 [M + H]^+^, 641.77 [M + Na]^+^, 657.87 [M + K]^+^.

(**12ab**) HPLC purity ≥ 99.0%; ^1^H-NMR (500 MHz, CD_3_OD, 25 °C, δ ppm, *J* Hz); 7.10 (d, 1H, H2′); 7.05–6.96 (overlapped signals, 3H, H6′, H5′, H2″), 6.89 (d, *J* = 7.7, 1H, H6″); 6.84 (d, *J* = 7.7, 1H, H5″); 6.75–6.67 (overlapped signals, 2H, H5‴, H2‴); 6.57 (d, *J* = 7.7, 1H, H6‴); 6.05 (s, 1H, H6); 6.00 (s, 1H, H8); 4.98 (dd, *J* = 11.5, *J* = 2.1, 1H, H2); 4.90 (overlapped signal with H_2_O, 1H, H7″); 4.50 (dd, *J* =11.5, *J* = 2.1, 1H, H3); 4.15–4.01 ( overlapped signals, 3H, H-8T, H8″); 3.86 (s, 1H, OCH_3_); 3.70 (d, *J* = 12.3, 1H, H9″a); 3.49 (dd, *J* = 11.4, *J* = 4.3, 1H, H9″b); 2.87 (t, *J* = 6.0, 2H, H7‴) ppm. ^13^C-NMR (125 MHz, CD_3_OD, 25 °C, δ ppm): 197.2; 167.6; 163.5; 162.7; 147.7; 146.9; 144.7; 144.0; 143.7; 143.5; 129.9; 129.3; 128.0; 120.8 (·2); 120.3; 119.9; 116.4 (·2); 116.2 (·2); 115.7; 115.0; 114.9; 110.6 (×2); 101.1; 95.2; 94.1; 83.3; 78.6; 76.3; 72.3; 69.3; 60.6; 55.0; 34.3 ppm. MS (MALDI-TOF) (+): calculated for [M] (C_33_H_30_O_12_) 618.17; found 618.55 [M + H]^+^, 641.67 [M + Na]^+^, 657.88 [M + K]^+^.

### 2.4. Synthesis of 7-O-tyrosyl-2,3-dehydro Silybin Derivatives (***13a***–***15a***): General Procedure

7-*O*-tyrosyl SilA derivatives **10a**–**12a** (0.08 mmol) were dissolved in 500 µL of DMF and 24 mg of KOAc (0.24 mmol) was added. The mixture was kept at 50 °C and after 45 min, the disappearance of the SilA derivative, followed by TLC (DCM/CH_3_OH/AcOH, 90:10:0.01, *v*/*v*/*v*) control, was observed. The crude material was purified by column chromatography eluted with DCM/CH_3_OH 90:10 (*v*/*v*) and the derivative 7-*O*-tyrosyl-2,3-dehydro silybin (**13a**–**15a**) was obtained in good yield (78–85%, see Table 1). RP-HPLC purification was carried out on a Phenomenex Gemini RP18 column (10 μm particle size, 21.20 mm × 250 mm i.d.) using a linear gradient of ACN in 0.1 M NH_4_OAc in H_2_O, pH 7.0, from 20% to 100% over 30 min at a flow rate of 8 mL/min, with detection at 288 and 260 nm. The purity of final products (**13a**–**15a**) was on average 99.0%.

(**13a**) HPLC purity ≥ 99.0%; ^1^H-NMR (500 MHz, DMSO-*d6*, 25 °C, δ ppm, *J* Hz); 7.82–7.81 (overlapped signals, 2H, H2′, H6′); 7.12–7.09 (overlapped signals, 3H, H5′, H3‴, H5‴), 7.04 (s, 1H, H2″); 6.88 (d, *J* = 7.7, 1H, H6″); 6.82–6.80 (overlapped signals, 2H, H8, H5″); 6.69 (d, *J* = 7.9, 2H, H2‴, H6‴); 6.31 (s, 1H, H6); 4.96 (d, *J* = 8.3, 1H, H7″); 4.27 (m, 1H, H8″); 4.22 (t, *J* = 6.6, 2H, H8‴); 3.78 (s, 3H, OMe); 3.56 (d, *J* = 11.7, 1H, H9″a); 3.36 (overlapped signal with H_2_O, 1H, H9″b) 2.93 (t, *J* = 6.6, 2H, H-7‴) ppm. ^13^C-NMR (125 MHz, DMSO-d6, 25 °C, δ ppm): 176.6; 164.7; 160.8; 156.6; 156.4; 148.1; 147.5; 146.5; 145.6; 143.9; 137.2; 130.4; 128.3 (×2); 127.7; 124.2; 121.9; 121.0; 117.3; 116.7; 115.8; 115.6 (×2); 112.1; 104.5; 98.3; 93.1; 79.0; 76.3; 69.8; 60.6; 56.2; 34.3 ppm. MS (MALDI-TOF) (+): calculated for [M] (C_33_H_28_O_11_) 600.16; found 600.38 [M + H]^+^, 623.44 [M + Na]^+^, 639.49 [M + K]^+^.

(**14a**) HPLC purity ≥ 99.0%; ^1^H-NMR (500 MHz, DMSO-*d6*, 25 °C, δ ppm, *J* Hz); 7.82–7.81 (overlapped signals, 2H, H2′, H6′); 7.12 (d, *J* = 9.8, 1H, H5′), 7.04 (s, 1H, H2″); 6.89–6.87 (overlapped signals, 2H, H6″, H2‴); 6.83–6.80 (overlapped signals, 2H, H8, H5‴); 6.69 (overlapped signals, 2H, H5″, H6‴); 6.33 (s, 1H, H6); 4.96 (d, *J* = 7.91, 1H, H7″); 4.28 (complex signal, 1H, H8″); 4.25 (t, *J* = 6.8, 2H, H8‴); 3.78 (s, 3H, OMe); 3.75 (s, 3H, OMe); 3.56 (d, *J* = 12.2, 1H, H9″a); 3.36 (overlapped signal with H_2_O, 1H, H9″b); 2.94 (t, *J* = 6.8, 2H, H7‴) ppm. ^13^C-NMR (125 MHz, DMSO-*d6*, 25 °C, δ ppm): 176.6; 164.7; 160.7; 156.6; 148.1; 147.9; 147.5; 146.5; 145.6; 145.5; 143.9; 137.1; 129.0; 127.7; 124.2; 121.9; 121.6; 121.0; 117.3; 116.7; 115.8 (×2); 113.6; 112.1; 104.5; 98.3; 93.1; 79.0; 76.3; 69.8; 60.5; 56.1; 56.0; 34.7 ppm. MS (MALDI-TOF) (+): calculated for [M] (C_34_H_30_O_12_) 630.17; found 631.66 [M + H]^+^, 653.56 [M + Na]^+^, 669.47 [M + K]^+^.

(**15a**) HPLC purity ≥ 99.0%; ^1^H-NMR (400 MHz, DMSO-*d6* +H_2_O, 25 °C, δ ppm, *J* Hz); 7.75–7.73 (overlapped signals, 2H, H2′, H6′); 7.06 (d, *J* = 8.55, 1H, H5′); 6.97 (d, *J* = 0.94, 1H, H2″); 6.83 (dd, *J* = 8.20, *J* = 1.41, 1H, H6″); 6.78–6.72 (overlapped signals, 2H, H5″, H8); 6.64 (d, *J* = 1.41, 1H, H2‴); 6.60 (d, *J* = 7.97, 1H, H5‴); 6.49 (dd, *J* = 8.03, *J* = 1.72, 1H, H6‴); 6.24 (s, 1H, H6); 4.89 (d, *J* = 7.98, 1H, H7″); 4.21 (complex signal, 1H, H8″) 4.14 (t, *J* = 6.27, 2H, H8‴); 3.72 (s, 3H, OMe); 3.50 (d, *J* = 11.4, 1H, H9″a); 3.30 (dd, *J* = 12.6, *J* = 4.6, 1H, H9″b); 2.80 (t, *J* = 6.48, 2H, H7‴) ppm. ^13^C-NMR (100 MHz, CD_3_OD, 25 °C, δ ppm): 176.1; 164.8; 160.7; 156.6; 147.9; 147.0; 145.8; 145.2; 144.8; 143.6; 143.5; 136.8; 129.5; 127.8; 124.2; 121.2; 120.3; 119.9; 116.5; 116.3; 115.8; 115.0; 114.9; 110.6; 104.0; 97.5; 91.9; 79.0; 76.3; 69.4; 60.6; 55.1; 34.4 ppm. MS (MALDI-TOF) (+): calculated for [M] (C_33_H_28_O_12_) 616.16; found 617.35 [M + H]^+^, 639.66 [M + Na]^+^, 655.87 [M + K]^+^.

### 2.5. Medium and Chemical Stability

The sample solution was prepared by dissolving the accurately weighed compound (**11a**) in DMSO and diluted with either RPMI1640 media supplemented with 100 U/mL penicillin, 100 mg/mL streptomycin, and 10% FBS or PBS at pH 7.4 to reach a final concentration of 100 µM (1% DMSO). The solution was placed at 37 °C in a heater. Samples of 0.2 mL were taken after t = 0, 0.5, 1 h, 3 h, 7 h, 24 h, and 48 h. The samples were treated with 0.2 mL of ice-cold ACN. Precipitated proteins were removed by centrifugation (SIGMA Laborzentrifugen GmbH, Osterode am Harz, Germany) with 10,000× *g* for 15 min and filtered. An amount of 80 µL of the solutions was directly analysed by the HPLC system (Shimadzu LC-9A, equipped with a Shimadzu SPD-6A Detector λ = 288 nm) using a RP18 column Phenomenex LUNA (5 μm particle size, 4.6 mm × 150 mm i.d.) eluted with NH_4_OAc 0.1 M with a linear gradient of 5–100% ACN in 20 min (flow = 0.8 mL/min).

### 2.6. DPPH Assay

The free radical scavenging activity of different concentrations of the test compounds was evaluated by their abilities to quench the stable 1,1-diphenyl-2-picrylhydrazyl radical (DPPH) in vitro. The DPPH solution (200 µM) was prepared in methanol and placed in the dark for 30 min before the analyses. The compounds were dissolved in methanol to prepare the stock solutions (1 mM–100 μM). DPPH solution was placed in test tubes (final concentration 50 µM), and the solutions of each compound (final concentration range 1–1000 µM) were rapidly added and mixed into every test tube to reach a final volume of 2 mL. The reaction was followed by a spectrophotometric analysis continuously measuring the absorbance at λ = 517 nm for 30 min. The percentage of inhibition (% inhibition) was calculated with the following equation:$$\%\;inhibition=\frac{A_{control}-A_{sample}}{A_{control}}\times 100$$

The EC_50_ value (the inhibition concentration of a sample at a 50% fall in absorbance of DPPH) was used to compare the DPPH scavenging activities.

### 2.7. ORAC Assay

The ORAC assay relies on free radical damage to a fluorescent probe, most commonly fluorescein, caused by an oxidizing reagent resulting in a loss of fluorescent intensity over time. Antioxidant protection can then be quantified by subtraction of the AUC (Area Under the kinetic Curve) of the blank reaction from those reactions containing antioxidants. The resultant difference is considered to be the antioxidant protection conferred by the sample compound. ORAC results are commonly reported as Trolox equivalents (TEs) calculated from comparison to a Trolox calibration curve. Briefly, 150 µL of the fluorescein solution (11.12 × 10^−2^ µM in phosphate buffer 0.75 mM, pH 7.4) was added into each well of a 96-well plate. Subsequently, 23 µL of buffer and 2 µL of stock solutions in DMSO of tested compounds were added to the wells to reach the final concentration range 1.25–20 µM. The plate was incubated for 30 min at 37 °C and then 25 µL of AAPH (2,2′-Azobis(2-methylpropionamidine) dihydrochloride) (152.6 mM) was added to each well. Immediately, the fluorescence was recorded by a microplate reader for 2 h in 1 min steps at 37 °C (λ_ecc_ = 485 nm, λ_em_ = 528 nm).

### 2.8. Cell Culture, Reagents, and Treatments

Prostate carcinoma PC-3 cells were purchased from the American Type Culture Collection (Manassas, VA, USA). PC-3 cells were maintained in RPMI-1640 medium, supplemented with 10% heat-inactivated foetal bovine serum, 100 U/mL penicillin G, and 100 µg/mL streptomycin sulphate from Thermo Fisher Scientific (Waltham, MA, USA). Cells were cultured at 37 °C in a humidified incubator with 5% CO_2_. Cells were initially plated and treated when they reached a confluency level of 70% to 80%. Cells exposed to varying doses of silybin or its derivatives (5 or 10 µM) were dissolved initially in DMSO. These treatments were administered for specific time intervals as described for each experiment. The concentration of DMSO in all treatments did not exceed 0.1% (*v*/*v*) in the medium.

The antibodies for cleaved caspase 3 (Asp175) (#9661) and β-Actin (#3700) were from Cell Signaling Technology (Beverly, MA, USA). Dimethyl sulfoxide (DMSO) was purchased from Sigma-Aldrich (St. Louis, MO, USA). Amersham™ ECL™ Western blotting detection reagents were from Fisher Scientific (Hampton, NH, USA).

### 2.9. Cell Growth and Death Assay

PC-3 cells were seeded in 35 mm plates at a density of 5 × 10^4^ cells per plate following the culture conditions described earlier. After a 24 h incubation period, cells were exposed to different treatments, including DMSO alone as a control or different silybin derivatives at 5 or 10 µM concentrations dissolved initially in DMSO for 48 and 72 h. Three separate plates were used for each treatment and time point. At 48 and 72 h post-treatment, adherent and suspended cells were harvested through trypsinization, centrifuged at 1200 rpm, washed with 1X phosphate-buffered saline (PBS), and placed into separate tubes. Each sample was counted in duplicate using a hemocytometer and an inverted microscope to ascertain the total cell count. The distinction between live and deceased cells was established by applying the previously described trypan blue dye exclusion method [38].

### 2.10. Flow Cytometry for Apoptosis and Cell Cycle Determination

We utilized an Alexa Fluor 488 Annexin V/PI kit from Invitrogen, Thermo Scientific (Waltham, MA, USA), to assess cell death. Cells were seeded at a density of 5 × 10^4^ cells per well in a 35 mm plate and treated with different derivatives of silybin, following the same protocol as in the cell growth assay, after a 24 and 48 h incubation period. The cell processing method described earlier was used to determine apoptotic cell death through flow cytometry. Flow cytometric analysis was conducted using a flow cytometer (NovoCyte Penteon Flow Cytometer, Agilent technologies, Santa Clara, CA, USA) within 30 min to quantify cells exhibiting Annexin V and/or PI positivity.

### 2.11. Lysate Preparation and Immunoblot Analysis

Cell lysates were prepared using a non-denaturing lysis buffer. To ensure uniform protein loading in each well, we employed the BCA method Bio-Rad (Hercules, CA, USA) to measure protein concentrations in the lysates. Samples containing 30–50 μg of protein per sample were then subjected to electrophoresis and transferred onto a nitrocellulose membrane. After blocking with a suitable blocking buffer, the membranes were exposed to a specific primary antibody for overnight incubation at 4 °C, followed by incubation with the appropriate peroxidase-linked secondary antibody and using ECL detection for visualization as described previously. Membranes were re-probed with an anti-β-actin antibody as a loading control.

### 2.12. Statistical Analysis

The statistical significance of differences between control and treated samples was calculated using an unpaired two-tailed Student′s *t*-test (GraphPad Prism 8.4; San Diego, CA, USA). A *p*-value of <0.05 was considered significant. Statistical significance was as follows: * *p* < 0.05, # *p* < 0.01, and $ *p* < 0.001 compared to control group scores.

## 3. Results

### 3.1. Synthesis, NMR Characterization, and Chemical Stability

Selective alkylation on silibinin has been extensively explored changing many parameters (different bases, solvents, and reagent equivalents) [22,39,40,41] and when these methods were extended to prepare silibinin alkylated derivatives [25,42], little or no yields were obtained especially due to the complete oxidation of silibinin to the 2,3-dehydro-silybin derivatives.

The challenge in our synthetic strategy grounds on the development of a mild and regioselective alkylation that avoids the oxidation of silybin into DHS. For this reason, we have chosen an alkylation via the Mitsunobu reaction [43]. The Mitsunobu reaction exploits the higher acidity of the OH in seven of the silybins and the higher nucleophilicity of the primary OH of the tyrosol units. Given the chemical nature of silybin consisting of different OH groups (one primary, one secondary, and three phenolic OH), we carried out the Mitsunobu alkylation starting from suitably protected building blocks. In this frame, to avoid self-alkylation by-products of silybins and to optimize the selectivity, we started from 9″-O-protected silybins and protected tyrosols at the phenolic OHs (Figure 1).

The *tert*-butyldimethylsilyl (TBDMS) has been chosen as the protecting group selectively inserted and removable under mild conditions, for both starting metabolites. The synthesis of 9″-OTBDMS, already reported by Křen et al. for silibinin (**1ab**) [16], was carried out by us starting both from silibinin (**1ab**), as well as from the two diastereoisomers silybin A (**1a**) and silybin B (**1b**).

Starting from the tyrosol-based metabolites **3**–**5** (Figure 1), only the phenolic functions have been protected by a recently fine-tuned approach. Briefly, all OH groups were protected with an excess of tert-butyldimethylsilyl chloride (TBDMSCl) in ACN/DMF (3:1, *v*/*v*) in the presence of Et_3_N. After regioselective deprotection of aliphatic OH group, by treatment with I_2_ (1% wt) in CH_3_OH, building blocks **6**–**8** were obtained in good yields (83–86%). For the synthesis of the 9″-OTBDMS silybins building blocks, silybin A or B (**1a** or **1b**) was reacted with TBDMSCl in ACN/DMF in the presence of pyridine obtaining, after suitable work-up and purification, **9a** or **9b** in 74% and 79% yields, respectively. Starting from tyrosol-based building blocks **6**–**8** and 9″-O- protected silybins **9a** and **9b**, and silibinin **9ab**, the 7-*O*-alkylation was set up by the Mitsunobu reaction.

In a typical Mitsunobu reaction, protected tyrosols (**6**–**8**) were reacted with 9″-OTBDMS silybins (**9a**, **9b** and **9ab**), in the presence of triphenylphosphine (PPh_3_) and diisopropyl azodicarboxylate (DIAD) in anhydrous THF at 0 °C. The crude reaction mixtures were found to be complex to purify so after simple chromatographic purification, they were treated with Et_3_N∙3HF in THF at room temperature. After RP-HPLC purification, the identity and complete signal assignment of compounds **10**–**12**, obtained in good yields (Table 1), was confirmed by 1D and 2D NMR analyses in combination with MS data. Table 2 summarizes the fully assigned ^1^H and ^13^C NMR data for compounds **10a**–**12a** based on the interpretation of COSY, HSQC, and HMBC spectra.

For example, in the case of **11a**, the signals in the ^1^H NMR and ^13^C NMR spectra confirmed the addition of a tyrosyl group to SilA (**1a**). By using heteronuclear multiple-bond correlation (HMBC), a long-range C-H experiment, the insertion of the tyrosyl group was assigned to the 7-OH of silybin. The key HMBC correlation found related the proton at δ_H_ 4.15 (CH_2_ 8‴ tyrosyl) to the carbon at δ_C_ 167.6 (C-7, Figure 2). Such correlation was also found in the HMBC spectra of derivatives **10a** and **12a**.

Although the Mitsunobu reaction was carried out starting from silybin building blocks with deprotected 3-, 5-, and 4″-OH groups, we did not observe the formation of alkylation products at the 5 or 4″ positions or the side reactions to lead to hydnocarpin-type products [44].

Subsequently, the treatment of derivatives **10**–**12** related to the **a**, **b,** and **ab** series with KOAc in DMF at 50 °C led to DHS derivatives **13**–**15** in good yields (Table 1). After HPLC purification, the identity of structures **13**–**15** was confirmed by 1D and 2D NMR analysis, in which we observed the disappearance of H-2 and H-3 silybin protons and the formation of the double bond C_2_-C_3_ at 146 and 137 ppm, respectively. In the case of silibinin derivatives **10ab**–**12ab**, some NMR signals, being a pair of diastereoisomers, appear as a pair and there are some overlapped signals. This issue does not exist with 2,3-dehydro-silibinin derivatives **13ab**–**15ab** since the NMR signals for both enantiomers are indistinguishable (see Appendix A).

The stability of 7-*O*-tyrosyl silybin derivatives has been investigated on **11a** as a representative compound. This investigation was accomplished over time by HPLC experiments, and the persistence was evaluated. The time-dependent stability in phosphate buffer at pH = 7.4 is not much different from that observed in the culture medium. We have observed a persistent presence (≥93%) of product **11a** even after 48 h. Noteworthy only after seven days, we observed the formation (~30%) of the silybin oxidation product, 2,3-dehydro-silybin (**14a**).

### 3.2. Radical Scavenger Properties (DPPH and ORAC Assays)

Recent studies on the *redox* properties and chelating abilities of silibinin have elucidated the role of the different hydroxyl functions and in particular the pro-oxidant role of the 7-OH function [29,45].

A preliminary antioxidant investigation of the new compounds was performed by the ORAC (Oxygen Radical Absorption Capacity) and DPPH (2,2-diphenyl-1-picrylhydrazyl) assays. The antioxidant activities are reported in Table 1 for all derivatives, along with tyrosol-based TYR, MTYR, and HTYR (**3**, **4**, and **5**), three well-known phenols, as well as silybins (**1ab**, **1a**, and **1b**) which are the reference flavonolignans. By the ORAC data, comparable antioxidant ability was observed for compounds **10**–**12** and their counterparts (**1ab**, **1a**, and **1b**) with compound **12** remained more active. For derivatives **13**–**15**, there is a decrease in activity when compared to their progenitors (**2ab**, **2a**, and **2b**).

Conversely, in the DPPH assay, a significant contribution of the tyrosol moiety was observed. All derivatives (**10**–**15**) show greater activity than their progenitors, with a significantly better free radical scavenging ability of 2,3-dehydro-silybin derivatives (**13**–**15**) compared to silybin derivatives (**10**–**12**). These results should be analysed considering the role of 7-OH in the antioxidant activity of starting compounds: silybins and 2,3-dehydro-silybins.

As known by the literature, 7-OH in silybins presents a pro-oxidant capacity; therefore, its masking led to increased antioxidant activity in the new derivatives, even when it was linked to a compound lacking antioxidant activity such as TYR. A further increase in scavenging activity has been observed for derivatives **11**–**12** where MTYR and HTYR are good antioxidants by themselves. Otherwise, in DHS, the 3-OH group is strongly involved in the radical scavenging activity of the metabolite. This behaviour was explained by several mechanistic studies carried out on selectively methylated silybin and DHS derivatives [29].

Accordingly, in DHS derivatives, the 7-OH was found to be crucial in the H atom abstraction mechanism of the 3-OH position by free radicals. The alkylation of 7-OH in DHS derivatives reduces the scavenging activity of the 3-OH group by decreasing the 3-OH bond dissociation enthalpy (BDE). In the ORAC assay, where the BDE plays an important role, there is an impairment of resonance stabilization of 3-OH and therefore a decrease in the antioxidant activity of alkylated DHS derivatives (**13**–**15**). On the contrary, the DPPH assay provides mainly an electron transfer mechanism (ET), so a synergic contribution of both DHS and the tyrosol moiety has been observed in the antioxidant activity of the resulting compounds.

### 3.3. Anti-Proliferative Effects towards Prostate Cancer Cell Lines

The trypan blue assay revealed that different silybin derivatives’ treatment at 5 and 10 μM concentrations for 48 and 72 h decreased the live cell number and induced cell death in PC-3 prostate cancer cells. Among all the silybin derivatives, compounds **15ab**, **15a**, and **15b** were found to be the most potent in restricting cell growth and inducing cell death in PC-3 cells at 48 and 72 h of treatment. Specifically, compounds **15ab**, **15a**, and **15b** revealed a significant reduction in the percent live cell count of PC-3 cells by ~5% to 41% (5 μM) and ~72% to 78% (10 μM) at 48 h (*p* < 0.001 for all, Figure 3A) and 33% to 62% (5 μM) and ~79% to 86% (10 μM) at 72 h (*p* < 0.001 for all, Figure 4A). Furthermore, the treatment of compounds **15ab**, **15a**, and **15b** of PC-3 cells significantly increased the percentage of dead cells by 1.6- to 2.5-fold (5 μM) and by 3.2- to 4.4-fold (10 μM) at 48 h (*p* < 0.01–0.001, Figure 3B) and by 1.6- to 2.6-fold (5 μM) and by 1.8- to 2.8-fold (10 μM) at 72 h (*p* < 0.05–0.001, Figure 4B) when compared to the control. The trypan blue assay results also revealed that compounds **14ab**, **14a**, and **14b** (Figure 3B and Figure 4B) also exhibited somewhat similar effects as compounds **15ab**, **15a**, and **15b** showed; however, the effect was prominent only at a higher concentration of 10 μM and at 72 h in case of compounds **14ab**, **14a**, and **14b**. Thus, for further studies, out of all the silybin derivatives investigated in the trypan blue assay, only compounds **15ab**, **15a**, and **15b** were used. Compound **1ab** was taken as parent control in further experiments.

### 3.4. Effects of 7-O-tyrosyl 2,3-dehydro-silybin Derivatives on PC-3 Apoptosis and Cell Cycle Progression

Prostate cancer PC-3 cells were seeded in 35 mm culture plates at a density of 5 × 10^4^ cells/plate. Following a 24 h incubation period, the cells underwent treatment with various concentrations (5 or 10 µM) of compounds **15ab**, **15a**, **15b**, and **1ab** for 24 and 48 h. Subsequently, the cells were harvested with trypsinization and then subjected to centrifugation (at 2500 rpm for 5 min), followed by staining with Annexin V and PI as per the manufacturer’s instructions. Results revealed that compounds **15ab** and **15b** were able to induce a significant increase in the late apoptotic cell population in PC-3 cells, which is positive for both Annexin V and PI staining. Specifically, a 10 µM concentration of compound **15ab** increased the late apoptotic cell population by 4-fold (*p* < 0.05, Figure 5B), and compound **15b** increased the late apoptotic cell population by 4.7-fold (*p* < 0.001, Figure 5B) at 24 h. Similarly, at 48 h, 10 µM of compound **15ab** increased the late apoptotic cell population by 3-fold (*p* < 0.001, Figure 5C), and a 10 µM concentration of compound **15b** increased the late apoptotic cell population by 3.5-fold (*p* < 0.001, Figure 5C) when compared to control. Results from Western blotting for cleaved caspase-3 also confirmed that compounds **15ab** and **15b** induce apoptosis in PC-3 cells. The expression for cleaved caspase 3 was significantly upregulated in 10 µM dose groups of compounds **15ab** and **15b** (Figure 6), thus confirming apoptosis induction by these specific silybin derivatives. Flow cytometric analysis for cell cycle progression revealed that compounds **15ab**, **15a,** and **15b** induced G1 phase arrest in PC-3 prostate cancer cells. Specifically, a 10 µM concentration of compounds **15ab**, **15a**, and **15b** increased the cells in the G1 phase by 1.4-fold, 1.2-fold, and 1.4-fold at 24 h, respectively (*p* < 0.01–0.001, Figure 7B). Similarly, a 10 µM concentration of compounds **15ab**, **15a**, and **15b** increased the cells in the G1 phase by 1.12-fold, 1.13-fold, and 1.14-fold at 48 h, respectively (*p* < 0.01–0.001, Figure 7C). 

## 4. Discussion

Redox homeostasis is essential for biological function; its imbalance leads to dangerous pathophysiological consequences in cells. The ROS levels are tightly regulated by antioxidants, and the most attractive are polyphenols that play a pivotal role in maintaining an optimal redox balance. With this premise, this work aims to modulate the antioxidant activity of the flavonolignan core fusing with other very attractive natural phenols without neglecting the stereochemistry of silybins through a regioselective synthesis of new 7-*O*-tyrosyl derivatives of Sil and DHS. New hybrid molecules were designed by combining the pharmacological properties of both silybins and tyrosol-based metabolites to improve both the antioxidant and anticancer activity of progenitors. Exploiting our knowledge of the orthogonal protection of different silibinin OH functions, we synthesized brand-new 7-*O*-tyrosyl silybin derivatives in very good yields and purity (Figure 1). Considering the capacity of the 7-OH hydroxyl group in triggering a pro-oxidant mechanism, this group has been derivatized with a tyrosol moiety by an ether bond. Alkylation of 7-OH of silybins has been carried out using 3-methoxytyrosol (MTYR) and 3-hydroxytyrosol (HTYR), known natural compounds with pronounced antioxidant and pharmacological properties [46,47].

Additionally, to outline a structure–activity relationship that considers not only the A and B stereochemistry of silybin but also the contribution of the group in the 7-*O* position, a very weak antioxidant such as tyrosol (TYR) has also been inserted. The synthesis of 7-*O*-tyrosyl derivatives was started from the pure diastereoisomers SilA **1a** and SilB **1b**, with silibinin (Sil, **1ab**) as a natural mixture. Oxidative treatment of the 7-*O*-tyrosyl silybin derivatives led to a new family of optically pure DHS derivatives (Figure 1). All derivatives, obtained with satisfactory yields, were fully characterized by 1D and 2D-NMR and MALDI-TOF analyses. For a model compound, the stability in PBS and culture medium was evaluated. Radical scavenger activity of all derivatives was evaluated employing various tests (DPPH and ORAC assays) and compared to that of silybin and tyrosol scaffolds. By a preliminary investigation of antioxidant properties, a very pronounced scavenging activity of some compounds against DPPH radicals was found. In particular, all data highlight the crucial role of the tyrosol moiety (**10** << **11** < **12**, of **a**, **b**, and **ab** series) while the stereochemistry of silybin A and B move into the background.

A significant contribution of the HTYR moiety on **12b** (5.65 ± 0.50 µM), slightly higher than that of **12a** (6.53 ± 0.60 µM), was observed, with much higher values if compared to those of the corresponding silybins (360, 580, and 620 µM of **1a**, **1b**, and **1ab**, respectively).

The same trend was observed for 2,3-dehydro-silybin-based derivatives with a much smaller difference (**13** << **14** < **15**, of **a**, **b**, and **ab** series).

Prostate cancer is the second most common cancer in the male population in Western society, and whereas there are options to manage the early stages of this malignancy, the advanced, aggressive, and androgen-independent stage of this cancer is mostly lethal. Clearly, one of the focusses to manage this stage of the disease is to utilize the tools that could be relevant to aggressive prostate cancer. Accordingly, the anticancer activity of all derivatives and relative mechanisms were investigated in the human prostate cancer PC-3 cell line; this selection, over the other human prostate cancer cell lines, was due to the heightened metastatic potential of PC-3 cells as compared to DU145 cells (moderate metastatic potential) and LNCaP cells (low metastatic potential). Furthermore, PC-3 cells are androgen-independent, making them a viable model for studying the aggressive form of prostate cancer resistant to androgen treatment. Given our study’s focus on investigating the efficacy of various silybin derivatives against this aggressive and androgen-independent prostate cancer, the utilization of PC-3 cells was considered appropriate. The anticancer activity of all derivatives and relative mechanisms were investigated in prostate cancer cells (PC-3 cell line). Prostate carcinogenesis is known to involve the dysregulation of the cell cycle, aberrant proliferation of cells, and evasion of apoptosis. The anticancer effects of silybin, in the past, have been attributed to its proliferation-inhibiting potential and the ability to induce cell cycle arrest and apoptosis [48]. The present study with 7-*O*-tyrosyl silybin and 2,3-dehydro-silybin derivatives also reveals similar results in PC-3 prostate cancer cells. In particular, **15ab**, **15a**, and **15b** were found to be the most potent of the library while all 7-*O*-tyrosyl 2,3-dehydro-silybin derivatives were able to significantly inhibit cell proliferation and induce cell death in PC-3 cells. The arrest of the G1 phase of the cell cycle, substantial increases in the late apoptotic population, and a marked enhancement in the expression of cleaved caspase 3 upon treatment with these compounds further supported their anticancer efficacy. Notably, it has been observed that some 2,3-dehydro-silybin derivatives, **15ab**, **15a,** and **15b**, exerted more potent anticancer activity than all synthesized derivatives in cell growth and death assays in human prostate cancer PC-3 cells. In-depth experiments such as flow cytometric analysis of the cell cycle and apoptosis as well as Western blotting experiments were carried out on the most active compounds.

## 5. Conclusions

New 7-O-alkyl silybin and 2,3-dehydro-silybin derivatives of the flavonolignan silibinin and tyrosol-based phenols (TYR, **3**; MTYR, **4**; and HTYR, **5**), were synthesized in good yields by regioselective Mitsunobu alkylation. New derivatives of SilA, SilB, as well as Sil have been synthesized starting from silibinin (**9ab**), silybins (**9a** and **9b**), and suitably protected tyrosol-based phenols (**6**, **7**, and **8**). All new silybin derivatives (**10**–**12**; **a**, **b**, and **ab** series) were subsequently oxidized by a simple and optimized one-step reaction, leading to the 2,3-dehydro-silybin derivatives (**13**–**15**; **a**, **b**, and **ab** series). A SAR profile has been outlined by evaluating antioxidant properties in the ORAC and DPPH assays, as well as anticancer activity using PC-3 cells.

The insertion of an HTYR moiety at the 7-OH position confers to both silybin and 2,3-dehydro-silybin an enhancement of the antioxidant capacity. This study reveals similarly promising anticancer activity on PC-3 prostate cancer cells. In the end compounds, **15ab**, **15a**, and **15b** were found to be the most potent of the library while all 7-*O*-tyrosyl 2,3-dehydro-silybin derivatives were able to significantly inhibit cell proliferation and induce cell death in PC-3 cells. These findings have a great impact since silibinin has already undergone clinical evaluations for its safety and anticancer properties. Consequently, the three more potent 7-*O*-HTYR 2,3-dehydro-silybin derivatives uncovered in this study warrant swift investigation to pre-clinical models and clinical trials. All compounds require a more complete investigation involving kinetic studies (IC_50_ data) and a screening on different prostate cancer cells. Moving forward, our research will encompass an evaluation of the in vivo anticancer efficacy of the promising derivatives, alongside an examination of their potential toxicity and bioavailability.

## Data Availability

Data are contained within the article and Appendix A.

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
