# Peer review of "7-O-tyrosyl Silybin Derivatives as a Novel Set of Anti-Prostate Cancer Compounds"

_antioxidants, 2024, doi:10.3390/antiox13040418_

Round 1

Reviewer 1 Report

This manuscript describes the synthesis of new Silybin derivatives using the Mitsunobu reaction. The reactions were well discussed and all the newly synthesized compounds described by many spectral data, were also reported in the supplementary material.

The introduction highlighted the aims of the authors. The main goal of this research has been achieved, the in vitro assays registered positive results, together with the antioxidant evaluation, that resulted in improvements after the chemical modification. The references are coherent with the whole manuscript and exactly addressed. The registered results can be considered of high scientific interest.

Page 1 line 3, write 7-O alkyl (O italic)instead of 7-O alkyl

Page 3 line 64:  the authors said: "Although silibinin (Sil, 1ab) does not present a significant antioxidant activity", this sentence must be discussed, exploring why this happens, despite there being many OH groups in the Sylibin scaffold.

Page 9 line 376 via instead of via

 Page 14 line 529: write 7-O tyrosyl instead of 7-O tyrosyl, the same thing on page 15 line 535, page 16 line 541, page 17 lines 548,  and 553. 567, 559, page 18 line 582, page 19 lines 594, 595, 596, 611

I suggest reducing the number of self-citation, 11 on 55

Author Response

Dear referee,

Let me thank you for the gratifying comments and constructive criticisms. I have found your remarks very useful to improve the quality of the manuscript, and all have been taken into account during this revision. Our point-by-point responses (as authors) to all reviewers’ comments (in italics) are reported below.

Major comments

This manuscript describes the synthesis of new Silybin derivatives using the Mitsunobu reaction. The reactions were well discussed and all the newly synthesized compounds described by many spectral data, were also reported in the supplementary material.

The introduction highlighted the aims of the authors. The main goal of this research has been achieved, the in vitro assays registered positive results, together with the antioxidant evaluation, that resulted in improvements after the chemical modification. The references are coherent with the whole manuscript and exactly addressed. The registered results can be considered of high scientific interest.

Detail comments

Page 3 line 64:  the authors said: "Although silibinin (Sil, 1ab) does not present a significant antioxidant activity", this sentence must be discussed, exploring why this happens, despite there being many OH groups in the Sylibin scaffold.

Authors response: As suggested, in the introduction section (Page 3, line 64) we discussed the redox properties of silibinin (yellow) in the revised manuscript. Silibinin is considered as pleiotropic acting compound since it’s not only able to handle as antioxidants, but in specific conditions also as pro-oxidant agent. The discrepancy between synergy or antagonism effects is well documented with some references.

Page 1 line 3, write 7-O alkyl (O italic)instead of 7-O alkyl

Page 9 line 376 via instead of via

Page 14 line 529: write 7-O tyrosyl instead of 7-O tyrosyl, the same thing on page 15 line 535, page 16 line 541, page 17 lines 548,  and 553. 567, 559, page 18 line 582, page 19 lines 594, 595, 596, 611

I suggest reducing the number of self-citation, 11 on 55

Authors response: All suggestions/corrections have been taken into consideration. In particular, the number of self-citations has been reduced (8/51).

Yours sincerely

Prof. Giovanni Di Fabio

Reviewer 2 Report

Dear Authors, Congratulations for the nice work, which could contribute in understand how 7-O-tyrosyl silybin derivatives can contribute to the prostate cancer treatment.

 I have a couple of major comments, in my opinion you should provide more information about prostate cancer, specially about the used cell line, in this regard I am wondering why you select PC-3 cells that share features with Prostatic small cell neuroendocrine carcinoma and not LNCaP cells that share common features with adenocarcinoma. when Adenocarcinoma has higher impact.   PSCN accounts for <0.5% to 1% of all prostate cancers.

The second comment why you don’t use a positive control? like a known antioxidant inhibitor? In my opinion is mandatory, but you can also justify your decision in the manuscript.

please see the above sections.

Author Response

Dear referee,

Let me thank you for the gratifying comments and constructive criticisms. I have found your remarks very useful to improve the quality of the manuscript, and all have been taken into account during this revision. Our point-by-point responses (as authors) to all your comments (in italics) are reported below.

Major comments

Dear Authors, Congratulations for the nice work, which could contribute in understand how 7-O-tyrosyl silybin derivatives can contribute to the prostate cancer treatment.

I have a couple of major comments, in my opinion you should provide more information about prostate cancer, specially about the used cell line, in this regard I am wondering why you select PC-3 cells that share features with Prostatic small cell neuroendocrine carcinoma and not LNCaP cells that share common features with adenocarcinoma. when Adenocarcinoma has higher impact. PSCN accounts for <0.5% to 1% of all prostate cancers.

Authors response: The authors express gratitude to the reviewer’s insightful comments. As suggested, we have added the following introduction/ rationale in the discussion section and highlighted the portion (yellow) in the revised manuscript. Prostate cancer is the second most common cancer in male population in Western society and whereas, there are options to manage the early stages of this malignancy, the advanced, aggressive, and androgen-independent stage of this cancer is mostly lethal. Clearly, one of the focuses investigated to manage this stage of the disease is to utilize the tools that could be relevant to aggressive prostate cancer. Accordingly, the anticancer activity of all derivatives and relative mechanisms were investigated in human prostate cancer PC-3 cell line; this selection, over the other human prostate cancer cell lines, was due to the heightened metastatic potential of PC-3 cells as compared to DU145 cells (moderate metastatic potential) and LNCaP cells (low metastatic potential) (see PMID: 1653586). Furthermore, PC-3 cells are androgen-independent, making them a viable model for studying the aggressive form of prostate cancer resistant to androgen treatment. Given our study's focus on investigating the efficacy of various silybin derivatives against this aggressive and androgen-independent prostate cancer, the utilization of PC-3 cells was considered appropriate.

The second comment why you don’t use a positive control? like a known antioxidant inhibitor? In my opinion is mandatory, but you can also justify your decision in the manuscript.

Authors response: The authors appreciate the reviewer's observation. It is pertinent to note that numerous studies have investigated silybin as the potent polyphenolic antioxidant renowned for its efficacy against many ailments including cancer. Many of these studies have utilized a known antioxidant positive control and demonstrated the strong antioxidant effects of silibinin (PMID: 12230878, PMID: 11520257, PMID: 27867187). However, the focus of our study was a comparative analysis between silybin alone and various 7-O-tyrosyl silybin derivatives. Our objective was to showcase that these derivatives are more effective than silybin alone in inhibiting the growth and progression of highly metastatic prostate cancer cells. Consequently, a known antioxidant positive control was not included in our study.

Does the introduction provide a comprehensive yet concise overview about the state of knowledge in the area of research? In my opinion, more information about prostate cancer in general and the used cell line must be provided.

Authors response: As suggested, we have added the introduction/rationale in the discussion section and highlighted the portion (yellow) in the revised manuscript.

Yours sincerely

Prof. Giovanni Di Fabio

Round 2

Reviewer 2 Report

Dear Authors, thanks for your corrections, but you forgot to highlight the corrections in the manuscript, never the less I found them!

You ca add the limitations of your work

Author Response

Dear referee,
thanks for your remark, I found it very helpful in improving the quality of the manuscript and took it into consideration during this revision.
Our manuscript reports preliminary data evaluating the activities of the new compounds synthesized with a yet unexplored strategy for alkylation at position 7. The latter was the driving force of our work.
A limitation of our work certainly concerns the number of cell lines tested. Tests will be in-depth on multiple prostate cancer cell lines with a kinetic approach (IC50 calculation). This was included in the manuscript in the conclusions.

Yours sincerely

Prof. Giovanni Di Fabio
